# Study of the Seismic Response on the Infill Masonry Walls of a 15-Storey Reinforced Concrete Structure in Nepal

**André Furtado \*, Nelson Vila-Pouca, Humberto Varum and António Arête** 

Construct-Lese, Departamento de Engenharia Civil, Faculdade de Engenharia da Universidade do Porto, 4200-001 Porto, Portugal; nelsonvp@fe.up.pt (N.V.-P.); hvarum@fe.up.pt (H.V.); aarede@fe.up.pt (A.A.)

\* Correspondence: afurtado@fe.up.pt; Tel.: +351-913307062

**Abstract:** Following the strong earthquake on April 25, 2015 in Nepal, a team from the University of Porto, in collaboration with other international institutions, made a field study on some of the most affected areas in the capital region of Kathmandu. One of the tasks was the study of a high-rise settle of buildings that were damaged following the earthquake sequence. A survey damage assessment was performed to a 15-storey infilled reinforced concrete structure, which will be detailed in the manuscript. Moreover, ambient vibration tests were carried out to determine the natural frequencies and corresponding vibration modes of the structure. The main aim of this manuscript is to present a numerical study concerning the influence of the masonry infill walls in the structure seismic response. For this, three numerical models were built discriminating the situations with and without damage and nondamaged infill walls. Validation and calibration of the numerical model was ensured by comparing the numerical frequencies with those obtained from ambient vibration tests. In addition, linear elastic analyses were carried out, using real accelerograms from the Gorkha earthquake to assess and quantify the major differences between the models in terms of inter-storey drifts ratios, inter-storey shear forces and seismic loadings.

**Keywords:** Nepal earthquake; high-rise reinforced concrete structure; masonry infill walls; ambient vibration test; survey damage assessment; numerical modelling

## 1. Introduction

Recent earthquakes demonstrated that the masonry infill walls have an important contribution in the seismic response of the reinforced concrete (RC) structures. Most of the structural and seismic codes consider the infill panels as nonstructural elements, which according to post-earthquake damages report and to several experimental studies is not correct. As a result, even light to moderate earthquake shaking/acceleration or drift levels can cause damage to the infill walls and this damage may result in life safety hazards, immediate evacuation and loss of function of buildings, limiting the use of internal spaces. In many cases, the influence of the infill panels showed to be the reason of extensive damages or even the buildings collapses [1,2].

Three failure mechanisms were observed in most of the analysed cases along the last major earthquake events and also after the Gorkha earthquake. The first is associated with cases where masonry walls do not extend towards all the inter-storey height for openings, leaving a short portion of the columns clear, creating a short-column mechanism. The second is associated also with the short-column mechanism but induced by the stair-slabs connected to the column. In both mechanisms, the non-consideration of the nonstructural infill panels, or of the secondary elements (as the staircases) in the design, may not represent the real behaviour of the columns, underestimating the column

stiffness and, consequently, of the forces attracted, leading to unexpected shear failure [3]. The failure of several structural elements was observed after the Gorkha earthquake due to this specific mechanism, according to some authors. Finally, the third and last one is related to the vertical stiffness irregularity due to the irregular distribution of the infill panels, which can lead to the concentration of the deformation in the storeys with less presence of these elements [4]. A high reduction of the infill panels on the ground floor is quite common in Nepal for commercial purposes or garages, which increase their seismic vulnerability. Several collapses were observed during the Gorkha earthquake. Finally, the local failure of the infill panel, characterized by the detachment of the panel from the envelope frame, diagonal cracking, shear sliding was also reported following the Gorkha earthquake, as well was observed in the last major earthquakes in Europe.

During the past three decades, the number of RC buildings in Nepal has increased considerably, mostly built by the landowners or local builders, too dependent on previous knowledge and experience which has led to insufficient detailing, bad quality of materials or lack of proper design rules and practice. The construction of RC buildings in Nepal presents several weaknesses on the quality control of materials (improper vibration of concrete, improper size of the aggregates and steel bars with insufficient ductility) and reduced construction quality (reinforcement detailing and provisions, and insufficient percentage of reinforcement), which have a direct impact on the bearing capacity as well as the deformation capacity of the structural elements. Another important issue regarding the seismic vulnerability of these types of structures is related to decreasing number of masonry walls on the ground floor, leading to the high potential to develop soft-storey mechanisms, and subsequent partial/total collapse of some buildings. However, it should be mentioned that the major part of the collapses and extensive damages that occurred in the Gorkha earthquake were related to masonry buildings and historic constructions [5].

Regarding the high-rise RC buildings' (with 10–18 storeys) performance from the earthquake of April 25, 2015, it was observed that most of the damage was related to the infill walls [6–8]. Figure 1 shows the structure damage of a 14-storey RC and from the observation of the vertical profile of the building, the damages are concentrated the first seven floors of the structure, as can be seen in Figure 1a. The damages are composed by detachment of the walls from the envelope frame (Figure 1b), diagonal cracking (Figure 1c) and some slight out-of-plane detachment of the wall from the frame. The reduced level of damage suits the proper seismic behaviour of these structures, which were designed as per the Indian National rules and standards. Even though, in many cases, damage was limited to the infill panels, it is worth noting that in many of these structures, the occupants had to be moved for temporary shelter (for some, more than a year) due to safety issues related to possible failure of the panels and until all repairs are complete. In some cases, the high costs associated with the repair of the buildings exceeded the structural costs of construction.

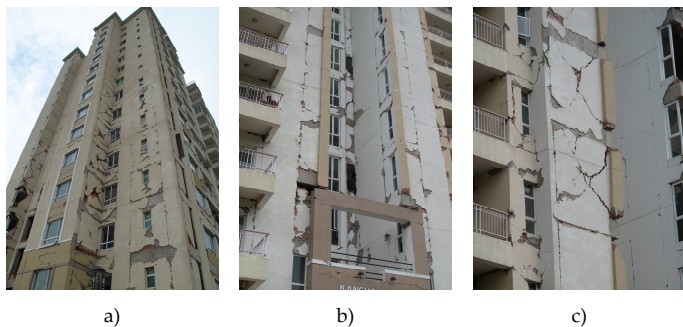

a)　　　　　　b)　　　　　　c)

**Figure 1.** Fourteen-storey reinforced concrete structure damaged after the Gorkha earthquake: (**a**) distribution of the damages from the first to the seventh storey; (**b**) detachment of the walls from the envelope frame; (**c**) diagonal cracking with slight out-of-plane detachment of the wall.

The main aim of the present manuscript is to study the effect of infill panels in the seismic response of a 15-storey infilled RC structure. For this, three numerical models were built discriminating the situations with and without damage and nondamaged infill walls. Comparing the numerical frequencies with those obtained from the ambient vibration tests ensured validation and calibration of the numerical model. In addition, linear elastic analyses were carried out, using real accelerograms from the Gorkha earthquake to assess and quantify the major differences between the models in terms of inter-storey drifts ratio, inter-storey shear forces and seismic loadings.

## 2. Case Study

### 2.1. Introduction

The study of the infill masonry walls influence in the seismic response of a RC structure was carried out based on the study of an existent structure in Nepal. The RC building is located in Kathmandu, is a 15-storey height, and is dated from 2012. Throughout the present section, the structure will be deeply described such in architectural or structural components. Posteriorly, the infill masonry walls characteristics and disposition will be detailed. An extensive damage survey assessment performed after the Gorkha earthquake focusing on the infill masonry walls' damages will be also included. Finally, ambient vibration tests were carried out to achieve the building structure vibration modes and corresponding natural frequencies. The major results of the testing campaign will be included within this section.

### 2.2. General Description

The building is a residential structure, located in the small town of Satdobato and belongs to a large luxury development composed of small houses and four high-rise structures (Figure 2a). The building is located at the left side of those grouped together. The structure is composed of two underground floors and 15 storeys (Figure 2b). The ground-floor height is 4 m and the others are 3.2 m high—a total of 48.8 m high.

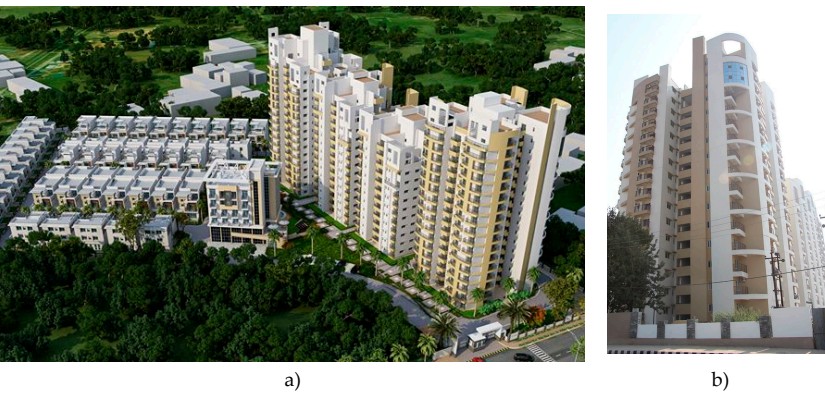

a)　　　　　　　　　　　　　　　　　　　　b)

**Figure 2.** Global overview of the high-rise structure under study: (**a**) location of the houses and the high-rise buildings; and (**b**) front view.

The tower is composed by RC frames filled with masonry infill walls made with solid clay bricks aligned according to the longitudinal and transverse direction and two stiff RC cores destined to the elevator boxes. The RC frames are composed by beam-column elements with spans relatively small. The beams are typically composed by two cross-sections, namely $300 \times 600$ mm and $230 \times 600$ mm. On the other hand, the columns are composed of 8 types of cross-sections, which are constant in height and very robust, as can be observed in Figure 3 and is described in Table 1. The longitudinal and transverse reinforcement is different among the different storeys. Complete detailing of all the cross-sections and reinforcements can be found in [9].

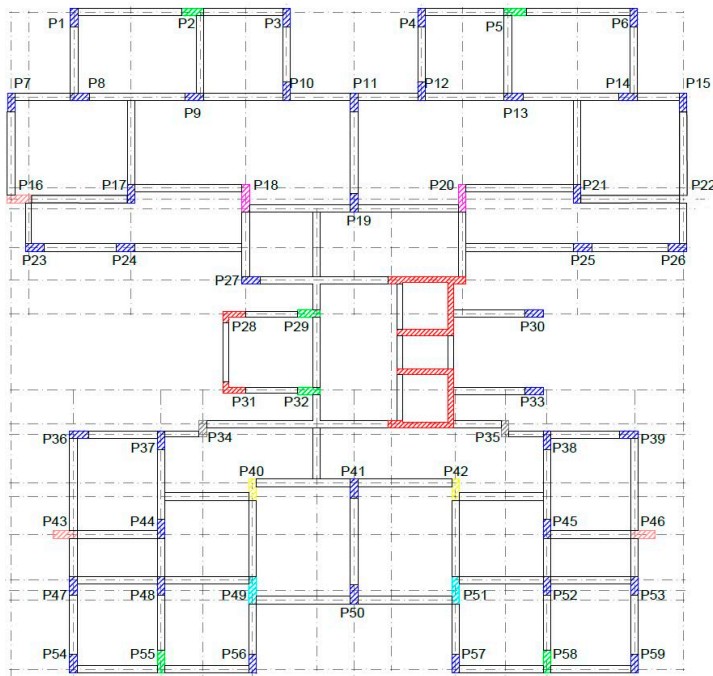

**Figure 3.** Case study: columns cross-section.

**Table 1.** Case study: columns cross-section and nomenclature.

| Dimensions [mm] | Color | Columns Nomenclatures |
|---|---|---|
| 900 × 300 | Fluorescent green | P2 ∣ P5 ∣ P29 ∣ P32 ∣ P55 ∣ P58 |
| 970 × 300 | Pink | P16 ∣ P43 ∣ P46 |
| 1125 × 300 | Hot pink | P18 ∣ P20 |
| 1100 × 300 | Baby blue | P49 ∣ P51 |
| 855 × 300 | Yellow | P40 ∣ P42 |
| 800 × 300 | Grey | P22 |
| 645 × 300 | Gray | P34 ∣ P35 |
| 750 × 300 | Blue | Remaining columns |

The solid RC slabs are supported by RC frame resisting system and have two different thicknesses: 125 mm and 110 mm. Two stiff RC cores destined to the elevator boxes are distributed in the middle alignment of the structure (red in Figure 3).

Concerning infill masonry walls, two different types of walls were noted from the visual inspection, namely: (i) Façade walls made with two leaf-panels, where each two rows of bricks are connected through one single row disposed perpendicularly, for a total thickness of 230 mm–250 mm; (ii) internal partition walls made with single rows of solid bricks with a total thickness of 150 mm. The solid bricks dimensions are 240 × 115 × 57 mm (length × thickness × height). A cement mortar was used for the horizontal and vertical bed joints, as well as for the plaster, which was approximately 2–3 cm thick.

Regarding the plan distribution of the infill panels, it was found the distribution was different between the ground floor and the remaining floors. At the ground floor, the major part of the walls did not exist to allow for the passage of people (Figure 4a). Through the observation of the plan disposition of the infill walls, some asymmetry is visible, which could introduce some torsion effect in the structure dynamic response. However, this could not be analysed without also considering the vertical structural elements, which also presents some asymmetry. Concerning the vertical disposition

of the infill walls seems to present also some asymmetry, since there is a high number of infill panels was observed in the top storeys, as can be seen in Figure 4b.

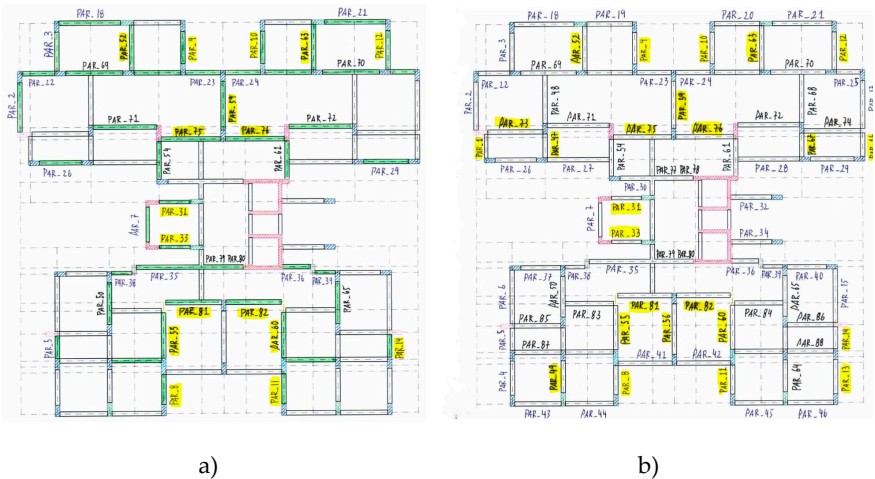

a)                                        b)

**Figure 4.** Distribution of the infill masonry walls: (**a**) ground floor; and (**b**) remaining floors. Where Par_€ means infill panel number €.

### 2.3. Damage Survey Assessment

Globally, the structural seismic performance was positive since the structural damages observed were slight to moderated. Figure 5 shows the vertical profiles of the structure from two different orientations, from which it becomes clear that the damage extension is higher in the bottom storeys (until 7th storey). Regarding the damages observed throughout the building's façades, only a few RC columns located in the ground floor suffered cracking and spalling of the concrete. It was not observed any column affected in the top storeys of the structure. Otherwise, the infill masonry walls suffered extensive damages. Diagonal cracking occurred in most of the cases followed by the detachment from the envelope frame. Out-of-plane collapse along the buildings' façade was not observed, which is justified by the robustness of the façade walls.

From the survey assessment of the interior of the building, it was observed that several panels detached from the surrounding frame and suffered diagonal cracking (Figure 6a,b). Due to the high stiffness of these infill panels, and a consequently reduced capacity to accommodate lateral distortions, shear sliding cracking was visible in some situations (Figure 6c). This damage was followed by the detachment of the panel from the surrounding envelope frame. Lastly, it was observed that some nonconfined interior partition walls suffered out-of-plane collapse (Figure 6d).

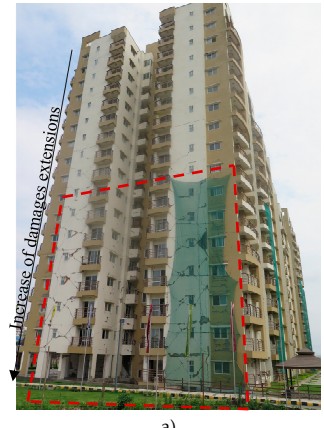
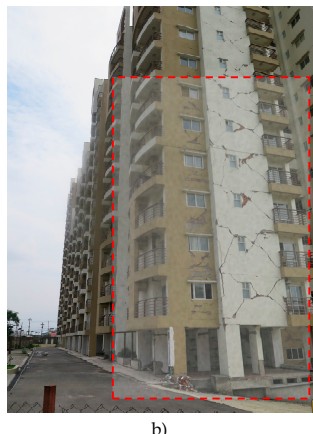

a)                                        b)

**Figure 5.** Survey damage assessment of the building: (**a**) north façade; (**b**) south-east façade.

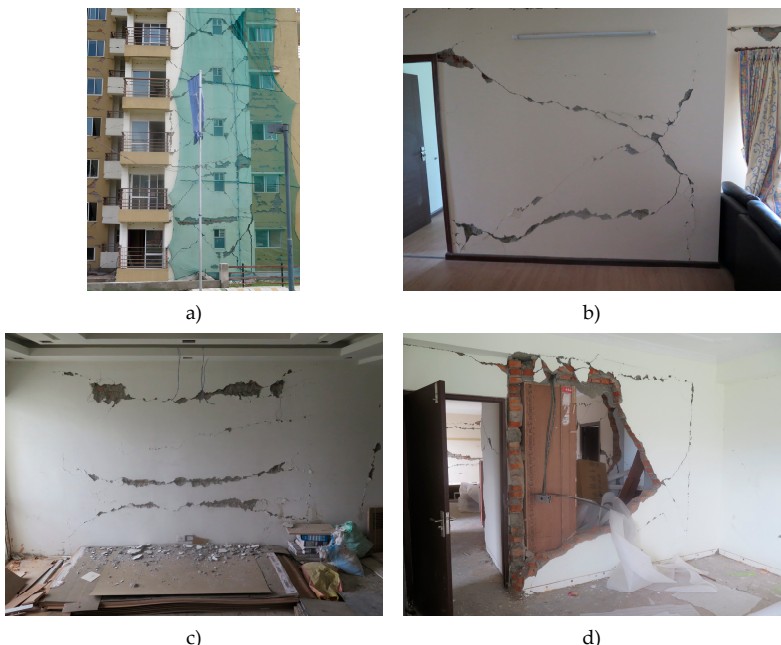

**Figure 6.** Survey damage assessment of the interior of the building: (**a**) diagonal cracking in façade wall; (**b**) diagonal cracking in interior partition wall; (**c**) sliding cracking in interior partition wall of storey 1; and (**d**) out-of-plane collapse of interior partition wall.

*2.4. Modal Identification Through Ambient Vibration Tests*

The modal identification combines experimental techniques with analytical methods characterization of the dynamic properties of a structure. It is often used to support the inspection and assessment of structures, when it is aimed to study the structural behaviour due to dynamic loadings such as wind or earthquake, or when it planned to determine the structural properties such as the lateral stiffness [10].

With the aim of obtaining the natural frequencies and the corresponding vibration modes of the damaged structure, ambient vibration tests were carried out. For this, three seismometers GeoSIG (GSR-18) were used. Each seismometer allows to recording acceleration signals of three orthogonal directions and assuming specific conditions of trigger, reading and sampling rates. Series of 900 s duration were considered, with defining sampling rates of 250 Hz. The adopted time series lengths for each setup were essentially limited by restrictions for the tests' duration; still, the presented results show that they were adequate for identifying of the most relevant natural frequencies. Each seismometer was set to work independently, avoiding the use of cables and minimizing the work associated with test preparation.

This was made possible by resorting to internal clock synchronization. The tests consisted in successive vibration measurements in building points, corresponding to different locations in the building plan (at the top story) and along the building height. The vibration data was collected through four different test setups, in which the seismometer number 3 (here designated S03) was the reference. This reference seismometer was placed in the center of the building plan, close to the elevators core at the 16th storey during all setups. In the first setup, seismographs S01 and S02 were placed on floor 16 in the left block of the building to capture the local torsion mode. The second test setup was comprised by two seismometers placed at two opposite corners (16th storey) aiming to capture the global torsion mode of the structure. Finally, in the third and fourth setup all seismometers were in the same plan location of the reference seismometer. The difference among the test setups was the position of the seismometers along the vertical height of the structure, namely in the setup 3 and 4, seismograph S01 and S02 were placed at the 10th storey and 13th storey, and the 4th storey and 7th storey, respectively. Figure 7 shows the floor layout for all setups.

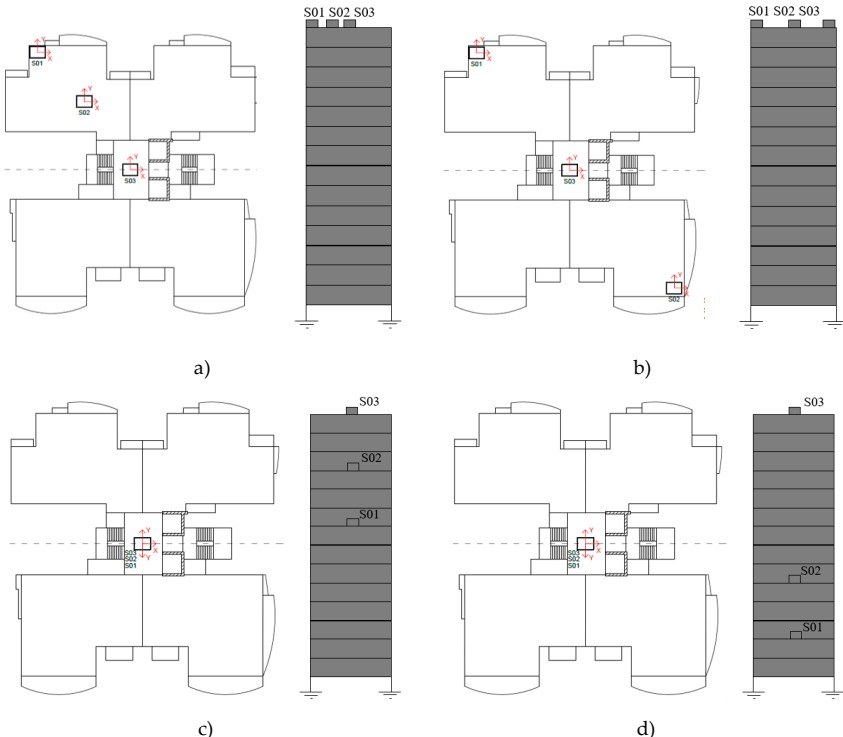

**Figure 7.** Ambient vibration tests: schematic layout of the seismometers disposition (**a**) Setup 1; (**b**) Setup 2; (**c**) Setup 3, and (**d**) Setup 4.

The determination of the natural frequencies and corresponding vibration mode shapes of the building is based on the acquisition of the acceleration measurements. For this purpose, the ARTeMIS [11] software for analysis and signal processing was used. The peak picking and the frequency domain decomposition (FDD) methods were used. Concerning the peak picking method, natural frequencies were identified from the peaks of the normalized average power spectra of the measured accelerations in each section, if the dynamic output in resonance is due only to one vibration mode shape. The modal identification was performed through the application of the enhanced frequency domain composition method (EFDD). The EFDD technique theory can be found in [12]. The spectral density matrices obtained from the analysis is plotted in Figure 8. From the plot 6 points are indicated which correspond to the structure natual frequencies. The 5-point red star is related to the first mode (translational mode along direction X—Figure 9a) equal to 0.61 Hz, and the blue one is related to the second mode (translational mode along direction Y—Figure 9b) equal to 0.75 Hz. Regarding the diamond scatter, it starts with the green one and it corresponds to a natural frequency equal to 1 Hz and is related to a torsional mode (Figure 9c). The gold diamond scatter corresponds to a second order vibration mode equal to 2.39 Hz (Figure 9d). Finally the gray and yellow diamond scatters correspond also to second order vibration modes equal to a natural frequency of 2.46 Hz and 2.78 Hz, respectively.

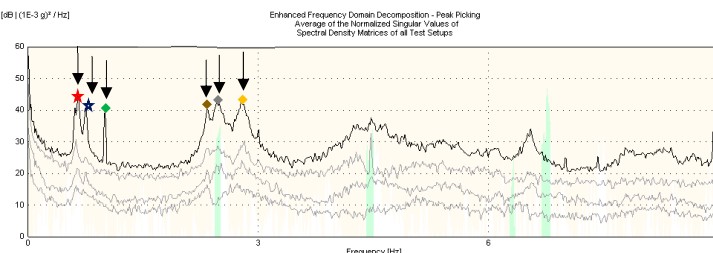

**Figure 8.** Ambient vibration test: normalized single values of spectral density matrices.

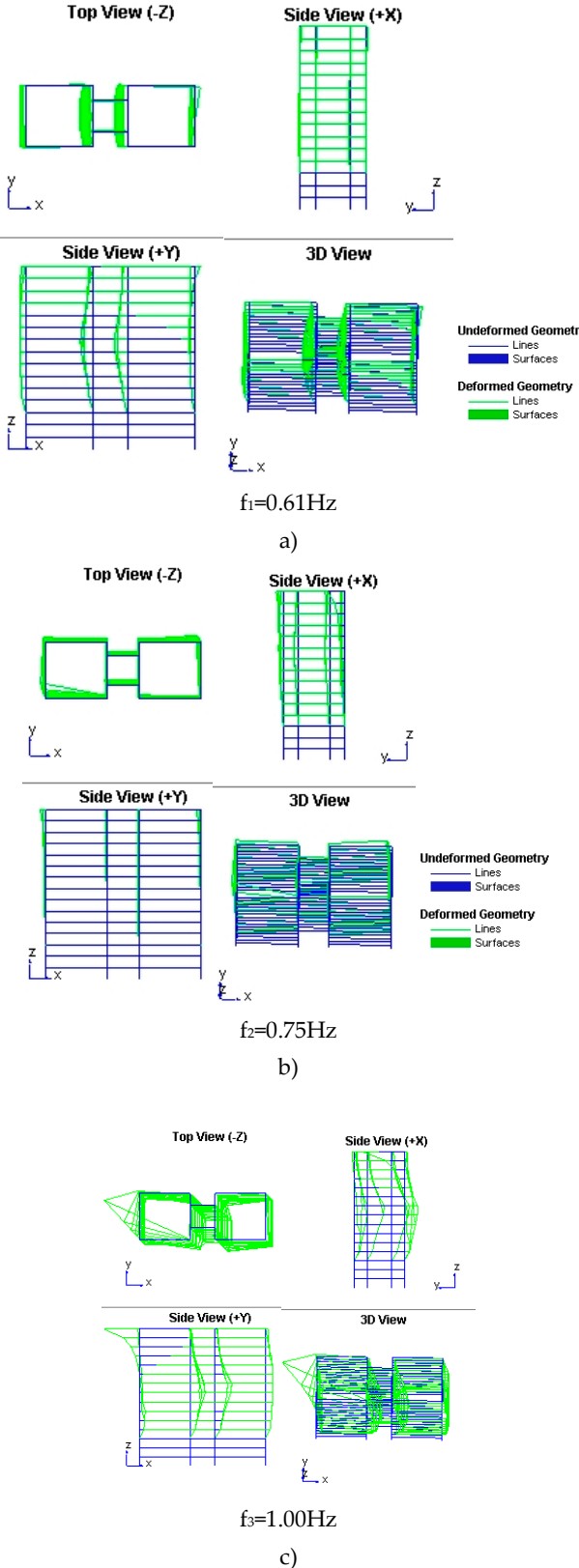

f₁=0.61Hz

a)

f₂=0.75Hz

b)

f₃=1.00Hz

c)

**Figure 9.** *Cont.*

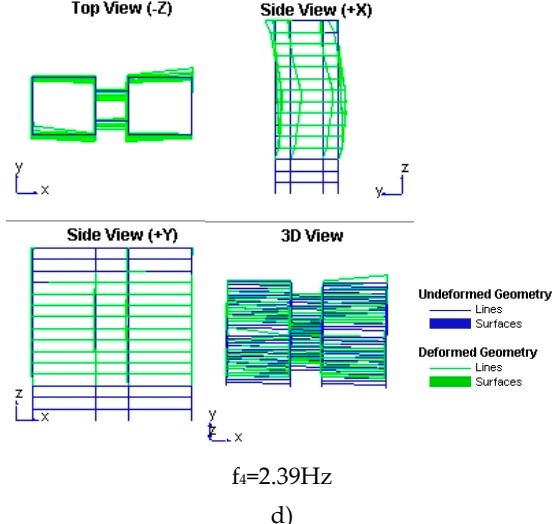

$f_4$=2.39Hz

d)

**Figure 9.** Ambient vibration test results: vibration modes corresponding to (**a**) 1st frequency; (**b**) 2nd frequency, (**c**) 3rd frequency and (**d**) 4th frequency.

## 3. Numerical Modelling

### 3.1. Introduction

The present section aims to detail the modelling strategies adopted within this work. Thus, it will start with the description of RC structural members modelling and then the modelling of the nonstructural elements. 3D numerical models were built in the software SAP2000 [13]. Three different numerical models were built considering different strategies related to the infill masonry walls, namely: (1) structure without infill masonry walls—Model 1; (2) structure with infill masonry walls (not considering existent damage)—Model 2; and (3) structure with damaged infill masonry walls—Model 3. The input material properties considered, as well as all the remaining modelling assumptions, will be discussed.

### 3.2. RC Structure Modelling

The numerical modelling of the building under study started by considering the plan and vertical disposition of the structural elements according to the structural drawings. Beams and columns were modelled through bar elements. Regarding the modeling of the two central RC cores, both were modelled using finite elements (plate elements) with dimensions that correspond to each thickness (Figure 10a). The slab modelling was performed by considering the rigid diaphragm at the storey levels, meaning each one behaved as rigid body (Figure 10b). Differences between infill masonry walls and rigid diaphragm concerning the dynamic action can be found in [14]. The modelling of the staircases was neglected since their contribution in terms of stiffness was considerably lower than the RC core's stiffness. However, the staircases' mass contribution was considered. A 3D view of the bare frame numerical model is shown in Figure 10c.

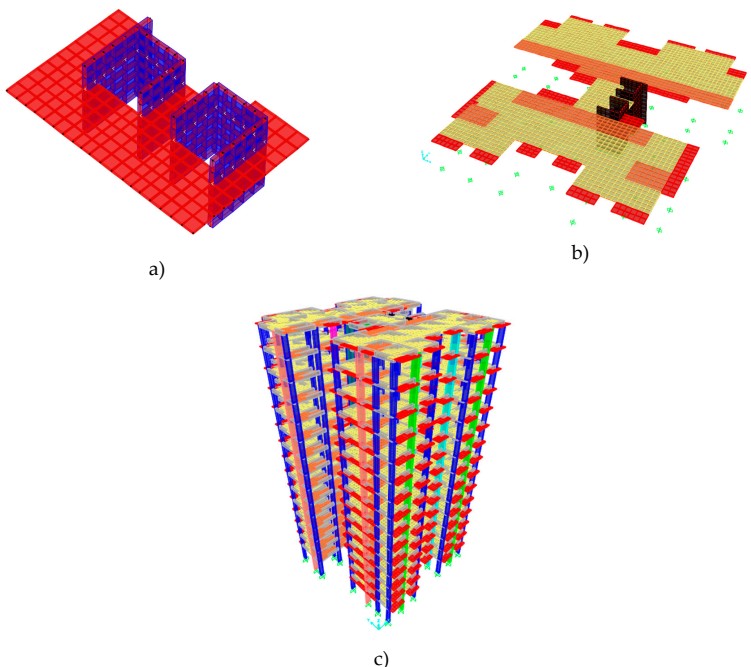

**Figure 10.** Numerical modelling of RC structure: (**a**) detail of the RC cores and slabs connection; (**b**) plan view; (**c**) 3D view.

Regarding the global vertical load, it was assumed a value of 5.09 kN/m$^2$ plus a variable load of 2.0 kN/m$^2$. The concrete compressive strength was assumed to be 20 MPa. A concrete elasticity modulus of 33 GPa and a tensile strength of 2.9 MPa was adopted.

*3.3. Infill Masonry Walls Modelling*

Many different approaches can be assumed to simulate the infill masonry walls seismic behaviour, starting from strut model concept [15–19] to detailed micro-modelling approaches [20]. Concerning the infill masonry walls modeling, the one-strut model approach was adopted, proposed by Al-Chaar [21] (Figure 11), which basically simulate the stiffness and strength contribution of the infills to the RC frame by the connection of the strut to the beam-column joints. Concerning the strut modelling parameters, the equivalent strut width, w, calculated for each infill panel was calculated according the Paulay and Priestley [22] proposal, which is given by Equation (1). The consideration of the infill panels' openings and quantification of the existent damage was achieved by the application of reduction factors according to the Al-Chaar [21] proposal. The authors suggested the application of the reduction factors $R_1$ and $R_2$ that affect the equivalent strut width. Thus, the effective reduced strut width, $w_{red}$, was calculated according to Equation (2). The reduction factor $R_1$ is related to the openings dimension, which is only applied if the panel area is lower than 60% of the panel area. For infill panels in which the openings area is higher than 60% of the panel area, some authors suggest to not consider the contribution of the wall in terms of strength and stiffness. The reduction factor $R_1$ was determined according to Equation (3).

$$w = 0.25 \times d \tag{1}$$

$$w_{red} = w \times R_1 \times R_2 \tag{2}$$

$$R_1 = 0.6 \times \left( \frac{A_{opening}}{A_{panel}} \right)^2 - 1.6 \times \left( \frac{A_{opening}}{A_{panel}} \right)^2 + 1 \tag{3}$$

Regarding the consideration of a reduction factor that considers the level damage, Al-Chaar [21] suggests the use of $R_2$, which can assume different values depending on the damage severity.

For example, in the case of a panel without damage, the coefficient $R_2$ is assumed as 1. Table 2 summarizes the $R_2$ reduction factor according to the panel geometry and level of damages.

Both simulate the infill masonry wall's in-plane behaviour through one equivalent strut (Figure 11).

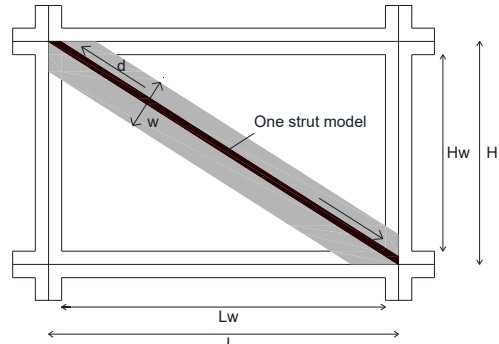

**Figure 11.** Infill masonry walls numerical modelling strategy: equivalent one strut model (w—strut width; $L_w$—panel length; $H_w$—panel height; H—inter-storey height; L—storey length).

**Table 2.** $R_2$ reduction factor.

| Hw/t | Moderated Damages | Extensive Damages |
|---|---|---|
| $\leq 21$ | 0.7 | 0.4 |
| $> 21$ | 0—Repairing strategies are needed | |

Two different numerical models were built with infill masonry walls: one without consideration of the damages and one considering the infill's damages. The quantification of the level of damage will be explained in Section 4, since the calibration was performed considering the natural frequencies.

## 4. Modal Analyses

### 4.1. Introduction

Throughout the present section, the modal analyses results will be presented and discussed. Starting from the model without infill masonry walls (Model 1) then the results from the model considering the infill masonry walls (Model 2) and, finally the results of the model considering the damaged masonry infill walls (Model 3). All the numerical model results will be compared with the experimental results; thus, the impact of the infill masonry walls presence in the structure vibration modes will be discussed. The comparison and discussion of the modal analyses results from the three numerical models will be presented in terms of natural frequencies and vibration modes of the structures.

### 4.2. Model Without Infill Masonry Walls—Model 1

From the modal analysis of the model without infill masonry walls it was collected the first four vibration modes and the corresponding natural frequencies and are summarized in Table 3.

**Table 3.** Modal analysis results: model without infill masonry walls (Model 1).

| Schematic Layout | Vibration Mode Deformed Shape | Natural frequency [Hz] |
|---|---|---|
|  |  | $f_1 = 0.52$ |
|  |  | $f_2 = 0.57$ |
|  |  | $f_3 = 2.14$ |
|  |  | $f_4 = 2.37$ |

*4.3. Model with Infill Masonry Walls (Not Damaged)—Model 2*

From the modal analysis of the model with infill masonry walls (not damaged), first four vibration modes and the corresponding natural frequencies were collected and are summarized in Table 4.

From the comparison between the numerical (Model 1 and Model 2) and the experimental frequencies, it can be observed that, as expected, that the frequencies of the model without infill masonry walls (Model 1) are quite lower than the experimental ones. The first and second frequencies are 15% and 25% lower, respectively. This is justified by the absence of infill masonry walls that results in the reduction of the global structure lateral stiffness. Regarding the model considering infill walls (Model 2), it is observed that the numerical frequencies are higher than the experimental ones. This can be accepted, since the experimental results are related to a structure with the panels damaged. The first frequency is 28% and the second one is 15%, respectively. A brief summary of the modal analysis results of the Model 1 and Model 2 is presented in Table 5.

**Table 4.** Modal analyses results: model with undamaged infill masonry walls.

| Schematic Layout | Vibration Mode Deformed Shape | Natural Frequency [Hz] |
|---|---|---|
|  |  | $f_1 = 0.78$ |
|  |  | $f_2 = 0.86$ |
|  |  | $f_3 = 2.52$ |
|  |  | $f_4 = 2.86$ |

**Table 5.** Modal analysis results: model with undamaged infill masonry walls.

| Type of Results | Natural Frequencies | |
|---|---|---|
| | 1st Vibration Mode | 2nd Vibration Mode |
| Experimental | 0.61 | 0.75 |
| Model 1 | 0.52 | 0.57 |
| Model 2 | 0.78 | 0.86 |

### 4.4. Model with Infill Masonry Walls (Damaged)—Model 3

The assignment of different levels of damage to the infill masonry walls was based in the Al-Chaar [21] proposal, and a reduction factor $R_2$ variation between 0.7 and 0.4 was considered according to the state of damage. Due to the high number of panels, it is very difficult to assume a

reduction factor for each panel. Thus, three different levels of damages were assumed based on the survey damage assessment (Figure 12).

**Table 6.** Modal analysis results: model with infill masonry walls (damaged)—Model 3.

| Schematic Layout | Vibration Mode Deformed Shape | Natural Frequency [Hz] |
| --- | --- | --- |
|  |  | $f_1 = 0.65$ |
|  |  | $f_2 = 0.77$ |
|  |  | $f_3 = 2.26$ |
|  |  | $f_4 = 2.56$ |

For the first level of damage, a reduction factor of 0.3 was assumed between the 1st and 6th storeys. For the second level of damage, between the 6th and 10th storeys, a reduction factor of 0.5 was assumed. Finally, for the third level of damage, a coefficient of 0.7 was assumed. From the modal analysis of the model with infill masonry walls (damaged), the first four vibration modes and the corresponding natural frequencies were collected and are summarized in Table 6.

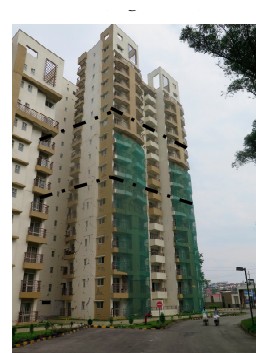
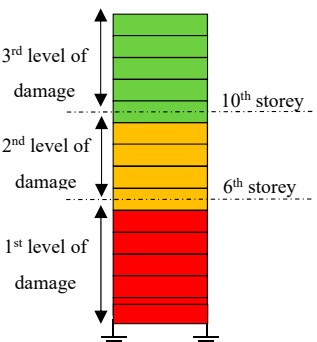

**Figure 12.** Schematic layout of the infill masonry wall level of damages assumed.

*4.5. Global Comparison*

From the global comparison of the results from the modal analyses, it can be noticed that the presence of the infill masonry walls increases the global lateral stiffness and thus the natural frequencies. However, the present study shows that the impact of the infill panel damages results in the reduction of the natural frequencies. At this point, the experimental results are important to discern which modelling strategy is more appropriate to simulate this structure. Table 7 summarizes the results from the modal analysis obtained by the three numerical modes and in the ambient vibration test. By the analysis of the first natural frequency (translational mode along direction X), the highest and the lowest value were achieved by Models 2 and 1 respectively. Model 2 is 28% higher than the experimental one and Model 1 is 15% lower. Concerning the comparison between the Model 3 and the experimental one, a small difference was obtained, namely 5% higher.

The second vibration mode, characterized by a translational mode along direction Y, similar results were found; the highest result was obtained by Model 2 with a frequency equal to 0.86 Hz (15% higher than the experimental one) and the lowest result was obtained by Model 1 (24% lower than the experimental one). Once again, Model 3 reached the result with higher accuracy, namely 0.77 Hz (2.6% higher than the experimental one). Similar observations can be drawn for the third vibration mode.

**Table 7.** Modal analyses results: global comparison.

| Model | Vibration Modes | | |
|---|---|---|---|
| | $f_1$ | $f_2$ | $f_3$ |
| Model 1 | 0.52 | 0.57 | 2.14 |
| Model 2 | 0.78 | 0.86 | 2.52 |
| Model 3 | 0.65 | 0.77 | 2.26 |
| experimental | 0.61 | 0.75 | 2.39 |

## 5. Linear Elastic Dynamic Analysis

*5.1. Introduction*

With the aim to assess the effect of the presence of infill walls in the structural response of the building, linear elastic dynamic analyses were carried out. Since this specific study is not related to the seismic vulnerability assessment of the structure, it was intended to perform only one dynamic linear elastic analysis with only one accelerogram for all the three numerical models. The selected accelerogram was the Gorkha earthquake, which is plotted in Figure 13. From the analysis of the spectral acceleration, it can be observed that the peak spectral acceleration occurs for natural periods between 0.24 s to 1.1 s. Looking for the natural periods of the numerical models, it seems that Model 2 (with undamaged infills) is the one closest to this range of natural periods (T = 1.25 s). This indicates that Model 2 will be the one subjected to higher seismic loading demands in this analysis.

For this analysis a stiffness reduction of the RC elements was considered according to the Greek code recommendations [23], which indicate reduction factors in structural analyses and assessment, and not the design of new structures. A reduction factor of 0.2 was considered for internal columns, 0.4 for external columns, 0.5 for cracked stiffness cores, 0.3 for noncracked stiffness cores and 0.6 for beams.

The analysis results will be analysed and discussed in terms of inter-storey displacements profiles, inter-storey drift ratios, inter-storey shear and inter-storey seismic loading. Four vertical alignments were analysed with the aim of accurately assessing the structural response (Figure 14).

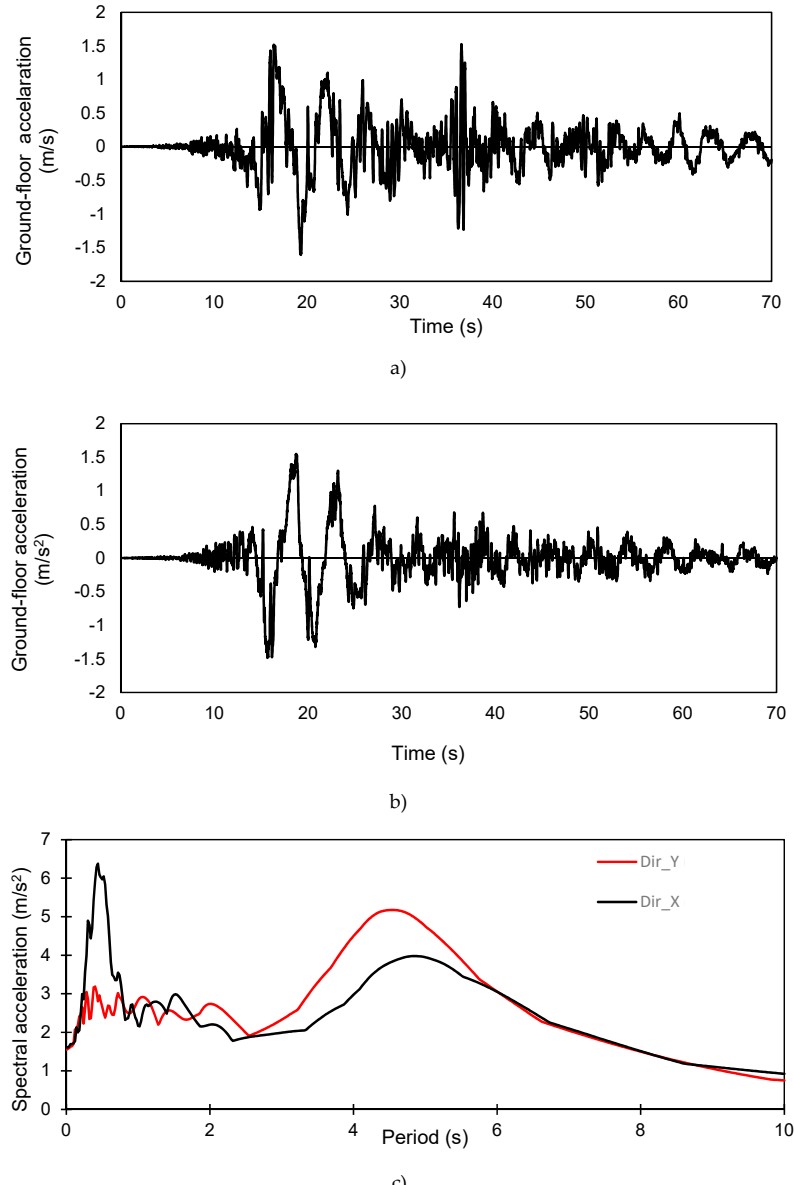

**Figure 13.** Linear elastic dynamic analysis: (**a**) accelerogram (direction X); (**b**) accelerogram (direction Y); and (**c**) spectral acceleration.

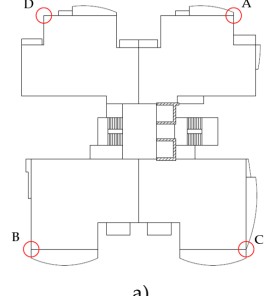
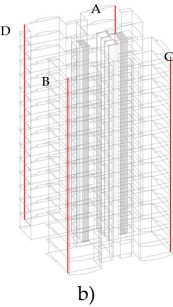

a)                                                                                                              b)

**Figure 14.** Linear elastic dynamic analysis: vertical alignments under study (**a**) plan view; (**b**) 3D view.

## 5.2. Inter-storey Displacements And Drift Ratio Profiles

Figure 15 presents the maximum inter-storey displacements of Model 3 along the four vertical alignments, from which it is easy to identify similar behaviour. It can be observed that the results from alignments A and D are similar as well as the results from the alignments B and C. It is possible to find that the maximum and minimum displacements in the alignments A and D occurred along direction Y, on the other hand in the alignments B and C occurred along the direction X. This difference can be justified by the torsional effect characterized by the geometry of this structure.

Based on this maximum and minimum inter-storey displacements, it can be also concluded that the A and D alignments along the direction X are 30% lower than the results obtained by the B and C alignments. The same does not occur for the inter-storey displacements along direction Y, where the results are quite similar. Once again this can be justified by the torsion effect.

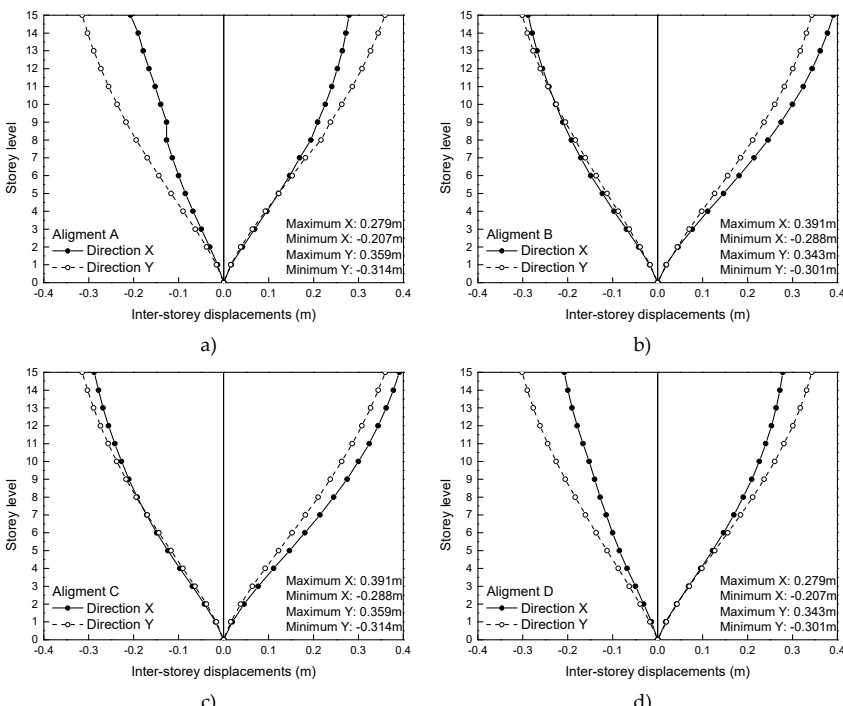

a)                                                                                                              b)

c)                                                                                                              d)

**Figure 15.** Linear elastic dynamic analysis result: maximum inter-storey displacement of Model 3 (**a**) alignment A; (**b**) alignment B; (**c**) alignment C and; (**d**) alignment D.

We also studied the level of the expected infills damage based on the relationship between the maximum inter-storey drift (ISD) ratio and the corresponding level of expected damage proposed by different authors. Based on the strut model, Magenes and Pampanin [24] proposed an empirical damage evaluation of the infill panels that corresponded to certain limit state, depending on the

axial deformation. FEMA-306 [25] and FEMA-307 [26] documents indicates also reference values of ISD ratio. The drift limit proposed for brick masonry is 1.5%, and the drift limit for the beginning of the diagonal cracking which is 0.25% can also be found in these documents, as can be observed in Figure 16.

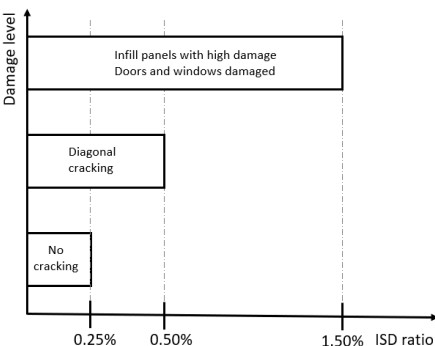

**Figure 16.** Relationship between maximum ISD ratio and infill masonry damage level proposed by FEMA-306.

From the analysis of the maximum inter-storey drift (ISD) ratio profiles (Figure 17), along the alignment A, the following observations can be drawn, namely:

(i)     In all the numerical models, in both directions, the higher ISD ratio values occurred in the intermediate storeys (between storeys 3 and 7);

(ii)    Globally, it can be observed that the higher ISD ratios occurred in the direction Y, with exception of Model 3 where it is observed the opposite. This can be justified by the irregularity of the damages observed in the infill masonry walls, which reduced the global lateral stiffness and strength and increased the lateral deformation;

(iii)   From the comparison between all the numerical models, it can be observed that Model 1 reached the highest ISD ratio values along the direction X, by achieving a maximum value of 0.85%. Model 2 reached the lowest one with a maximum ISD ratio of 0.43%. Concerning the response of Model 3, the impact of the infill panel damages increased the ISD ratio to a maximum value of 0.72%, which is 15% lower than the value obtained by Model 1 (Figure 16);

(iv)   Concerning the direction Y (Figure 16), different results were found since Model 2 and Model 3 reached similar response by reaching a maximum ISD ratio around 0.57%. On the other hand, Model 1 reached again the maximum ISD with a value of 0.83%.

(v)    Regarding the drift limits, it is possible to observe that along the damages are concentrated in the storeys above the 8th; most of them had diagonal cracking as observed in situ and reported in Section 2.3. The numerical results are quite similar—from the ISD ratio envelope of Model 2 and the storeys that exceeded the drift limits, as well as in comparison with the observed post-earthquake damages.

A similar trend was observed in the alignment C, however higher ISD ratios were achieved. Starting from the results of Model 1 (Figure 18a), the maximum ISD reached in the direction X is 26% higher than the one in direction Y. The storeys that obtained high level of ISD were the storeys between 2 and 8.

Concerning Model 2 (Figure 18b), lower differences were noticed in both directions. In fact, a small variation of around 5% is observed with the highest one along direction Y. In this case, it is not visible a special concentration of the drift demand along the building height.

Finally, from the analysis of the Model 3 response, the higher level of ISD ratios along the direction X is again visible, which reached a maximum value of 77% higher than the one reached in direction Y. The higher levels of drift occurred along the storeys 2 until 9, with storey 4 having the maximum ISD.

From the global analysis of the ISD ratios, a similar trend is once again visible, namely in the direction X the Model 1 and 2 achieved the highest and the lowest values (Figure 18b), respectively. Model 3 reached a maximum drift of 0.73%, which is 38% lower than the one reached by Model 1.

Concerning direction Y, once again Model 1 reached the maximum ISD ratio followed by the Models 2 and 3, respectively. The maximum value was around 0.92%, which is 39% and 42% higher than the Models 2 and 3, respectively as observed in the alignment A results.

Finally, regarding the drift limits exceedance it is possible to draw the same observations performed for the alignment A, which basically shows that the storeys between the 2nd and 8th storeys were most affected by the earthquake motion.

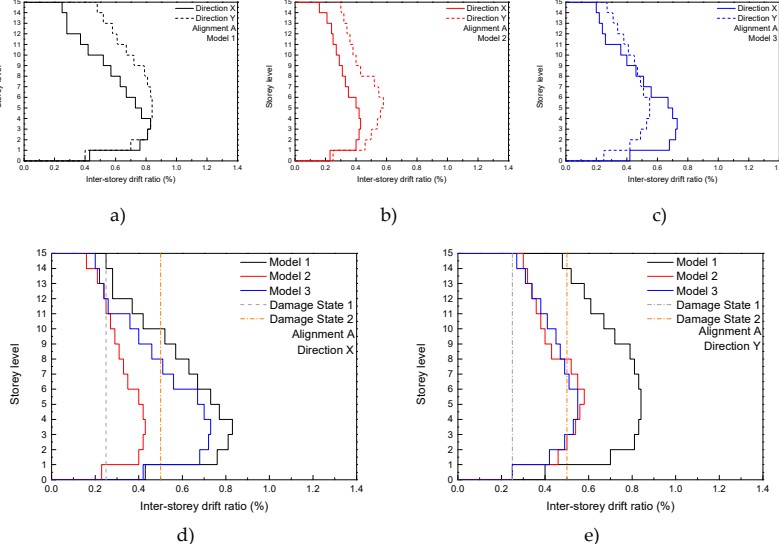

**Figure 17.** Linear elastic dynamic analysis result: maximum inter-storey drift ration (alignment A) (**a**) Model 1; (**b**) Model 2; (**c**) Model 3; (**d**) global comparison (Direction X); and (**e**) global comparison (direction Y).

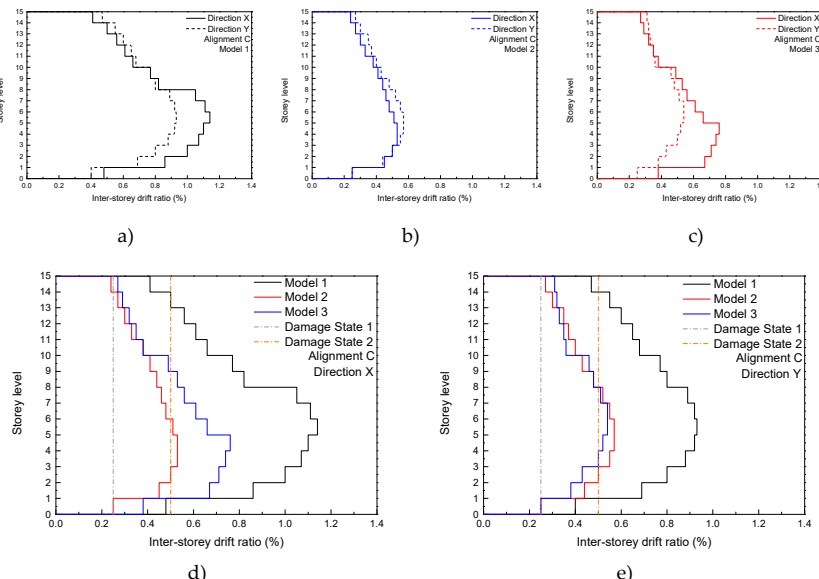

**Figure 18.** Linear elastic dynamic analysis result: maximum inter-storey drift ration (alignment C) (**a**) Model 1; (**b**) Model 2; (**c**) Model 3; (**d**) global comparison (Direction X); and (**e**) global comparison (direction Y).

### 5.3. Inter-storey Shear and Seismic Loading Envelopes

Concerning the analysis of the inter-storey shear, the maximum shear reached in each storey was extracted from the results, which are plotted in Figure 19. From the results it is possible to observe that the higher values were reached along the direction X (except Model 3). From the plots, it is visible that in Model 1 reached lower maximum storey shear variation along the storey height (Figure 19a). Otherwise, Model 3 was the one where a larger difference was noticed (Figure 19c). Model 2 appears to be an intermediate situation, such in direction X or Y (Figure 19b). From the global comparison along direction X (Figure 19d), it is possible to observe that Model 3 achieved results about 15% higher than Model 1 and 8% more than Model 2. The presence of the infill walls resulted in the increment of the storey shear, which could lead to significant impact on the frame columns and/or beam-column joints. Concerning direction Y (Figure 19e), it can be observed that the maximum values were reached by Model 2, followed by the Model 1 and then Model 3. Model 1 results were about 23% higher than that of Model 3, which is very similar to the model without infill walls.

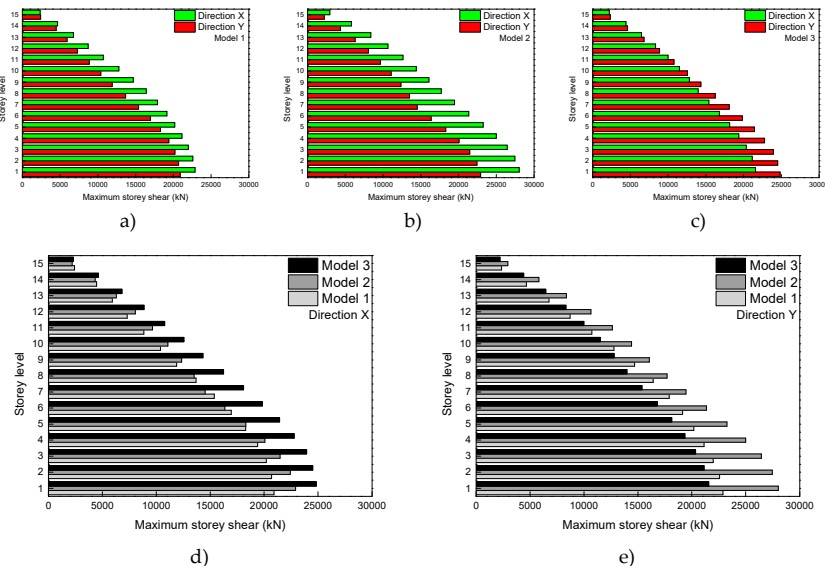

**Figure 19.** Linear elastic dynamic analysis result: maximum storey shear (**a**) Model 1; (**b**) Model 2; (**c**) Model 3; (**d**) global comparison (Direction X); and (**e**) global comparison (direction Y).

Finally, concerning the maximum seismic loading (Figure 20), it is visible that once again the higher values can be found along direction X in Model 1 and in Model 2. In other hand, Model 3 reached the higher maximum seismic loading along direction Y. By analysing the envelopes, it is possible to observe that in Model 1, the maximum seismic loading occurred for the storeys 8, 7 and 6 and the maximum ones in the 14th and 15th storeys (Figure 20a). Higher differences among storeys are visible in Model 2 (Figure 20b), namely the maximum values were reached along Storeys 5, 6 and 4 and finally in 14 and 15. Finally, in Model 3 (Figure 20c), the maximum seismic loadings were reached in the 8th, 7th and 9th storeys and finally in the top storeys (15 and 14).

From the global comparison between the numerical models it seems that along the direction X from the bottom until the 6th storey that Model 2 reached higher seismic loadings. From the 6th storey until 14th, Model 3 exceeded Model 2's seismic loadings. On the 15th storey, Model 1 achieved the highest value (Figure 20d). Finally, along direction Y, it seems that Model 2 always reached the highest seismic loadings when compared with the remaining ones. From the plot (Figure 20e), it is visible that the maximum seismic loadings were about 20% and 10% higher than Model 1 and 3 respectively.

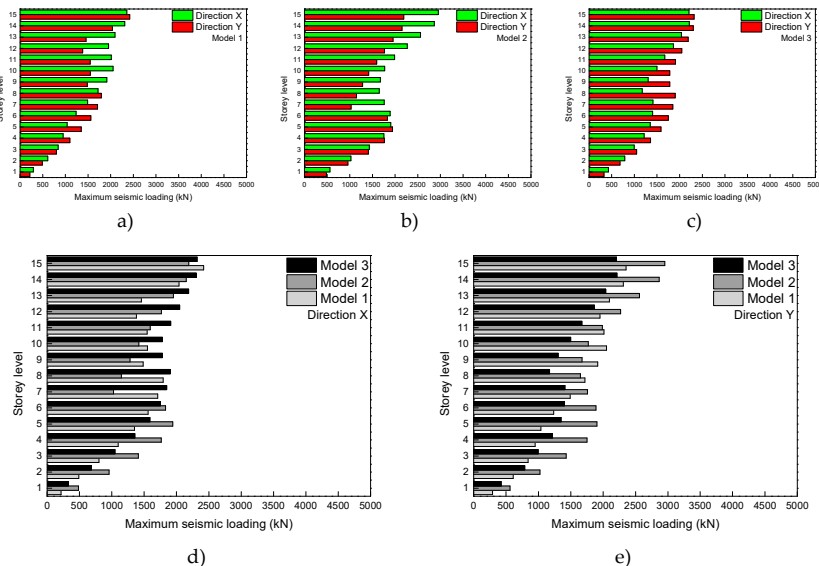

**Figure 20.** Linear elastic dynamic analysis result: maximum seismic loading envelope (**a**) Model 1; (**b**) Model 2; (**c**) Model 3; (**d**) global comparison (Direction X); and (**e**) global comparison (direction Y).

## 6. Conclusions

The main aim of this manuscript was to study the impact of the infill masonry walls presence in the seismic performance of a 15-storey high-rise building located in Nepal. The building was subjected to the Gorkha earthquake sequence in 2015 and was visited by an international team that performed a complete damage survey assessment report (herein presented). From the damage assessment, it was concluded that the major part of the observed damages found were related to the presence of infill masonry walls, namely local failure such as diagonal cracking, shear sliding cracks and detachment of the panel from the envelope frame. It was concluded that due to the high flexibility of the RC structure, the infill walls were subjected to significant deformations, which resulted in the observed extensive damages. Due to the seismic design of the RC structural elements, significant damages within the structural elements were not observed.

Ambient vibration tests were carried out to collect the vibration modes and the corresponding natural frequencies of the building under study. From the results, it was visible that the first and second vibration modes are characterized by slight torsion, which is due to the building geometry. The results obtained were used to calibrate the numerical model built in the software SAP2000. Additionally, two different numerical models were also built, considering different modelling strategies related to the infill masonry walls (without infill walls and with undamaged infill walls).

From the modal analysis, it was possible to observe that the infills presence increased the frequencies about 30%. Any modification of the of the vibration modes due to the infill's presence was not visible. From the comparison between the models with undamaged and damaged infill panels, it was found that neglecting the panel damage could result in differences between 10–20%.

Linear elastic analyses were carried out to assess the impact of the infill panels in the expected dynamic response of the structure. From the results, it was observed that the infill panels presence increased significantly the storey shear, and the maximum base shear about 20%. This important increasing of shear loadings due to the infills' presence is very important, since in the case of structures designed only to support gravity loads, it should be analysed carefully if the building' foundations and vertical elements are capable to support those shear loadings. From the results, it can be also concluded that the presence of the infill walls reduced the ISD ratio, however when compared with the drift limits proposed by FEMA-306, it was observed that a large part of the infill panel's damages occurred above the 8th storey. It was also observed that the presence of infill masonry walls contributed to the increasing of the torsion effect. From this study, it is evident that the infill masonry

walls played an important role in the seismic performance of the structure, highlighting the need to consider these elements during the design of new structures and/or the structural safety assessment of existing structures.

**Author Contributions:** Conceptualization, A.F. and N.V.-P.; methodology, A.F. and N.V.-P.; software, A.F., validation, A.F. and N.V.-P; formal analysis, A.F.; investigation, H.V. and A.A.; resources, H.V. and A.A.; writing—original draft preparation, A.F.; writing—review and editing, A.F. and N.V.-P; visualization, A.F.; supervision, N.V.-P., H.V. and A.A. and P.D.; project administration, H.V. (PDF) Seismic Performance of High-Rise Condominium Building during the 2015 Gorkha Earthquake Sequence. Available from: https://www.researchgate.net/publication/330673852_Seismic_Performance_of_HighRise_Condominium_Building_during_the_2015_Gorkha_Earthquake_Sequence [accessed Feb 02 2019].

**Funding:** UID/ECI/04708/2019—CONSTRUCT—Instituto de I&D em Estruturas e Construções funded by national funds through the FCT/MCTES (PIDDAC) and P0CI-01-0145-FEDER-016898—ASPASSI—Safety Evaluation and Retrofitting of Infill masonry enclosure Walls for Seismic demands.

**Acknowledgments:** This work was financially supported by: UID/ECI/04708/2019—CONSTRUCT—Instituto de I&D em Estruturas e Construções funded by national funds through the FCT/MCTES (PIDDAC) and by national funds through FCT—Fundação para a Ciência e a Tecnologia, namely through the research project P0CI-01-0145-FEDER-016898—ASPASSI—Safety Evaluation and Retrofitting of Infill masonry enclosure Walls for Seismic demands. The authors also acknowledge the constructive comments and suggestions given by the anonymous reviewers that improved the quality of the manuscript.

**Conflicts of Interest:** The authors declare no conflict of interest.

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
