# Peer review of "Study of the Seismic Response on the Infill Masonry Walls of a 15-Storey Reinforced Concrete Structure in Nepal"

_buildings, doi:10.3390/buildings9020039_

Round 1

Reviewer 1 Report

In the Reviewer opinion the paper is correct. Some comments which greatly enhance the understanding of the paper and its value are presented below. Specific issues that require further consideration are:

The title of the manuscript is matched to its content.

The structure of the manuscript is proper.

The Introduction cover the cases.

In the Reviewer’s opinion, the current state of knowledge relating to the manuscript topic has been covered and presented.

The presentation of the results are correct.

Conclusions bring important scientific and cognitive elements.

In the Reviewer’s opinion, the bibliography is representative and exhaustive.

In the Reviewer’s opinion the manuscript can be published in the Journal.

Author Response

After carefully reading of the reviewers’ comments, the authors have considered all the changes suggested, and in the following sections the comments/revisions of each reviewer and the corresponding answers/explanations made by the authors will be presented. The authors really appreciated the work done by the reviewer who they want to acknowledge for the useful comments and remarks.

The authors would like to acknowledge the revision and the comments of the reviewer. The manuscript was slightly improved and the manuscript language was also revised.

Reviewer 2 Report

Dear Authors

the paper is potentially of high interest but needs major and important corrections. The requested improvement/corrections are reported in shynthetic way as following:

- The introduction needs to be improved and rewritted since that the influnce due to infill masonry walls in normal and taller framed building subjected to horizontal dynamic action is well known especially in Europe.

- The introduction has to be improved with fruitfuill comparison than masonry bearing cracked part damaged in Nepal after the same earthquake in historic constructions

- The improve the introsduction also the following new reference are suggested:

1.  Brando G., Rapone D., Spacone E., O’Banion M. S., Olsen M. J., Barbosa A. R., Faggella M., Gigliotti R., Liberatore D., Russo S., Sorrentino L., Bose S., Stravidis A. (2017). Damage Reconnaissance of Unreinforced Masonry Bearing Wall Buildings After the 2015 Gorkha, Nepal, Earthquake. EARTHQUAKE SPECTRA, vol. 33, p. 243-273, ISSN: 8755-2930, doi: 10.1193/010817EQS009M

2. Russo S. (2016). Integrated assessment of monumental structures through ambient vibrations and ND tests: The case of Rialto Bridge. JOURNAL OF CULTURAL HERITAGE, vol. 19, p. 402-414, ISSN: 1296-2074, doi: 10.1016/j.culher.2016.01.008

3. Dal Cin A., Russo S. (2016). Annex and rigid diaphragm effects on the failure analysis and earthquake damages of historic churches. ENGINEERING FAILURE ANALYSIS, vol. 59, p. 122-139, ISSN: 1350-6307, doi: 10.1016/j.engfailanal.2015.09.010

The authors are not forced to use the new above proposed references in the introduction but they can concretely improve that chapter and the value of the references. By the way, with the aim to facilitate their use in the Introduction, the reference 1 gives the possibility to better describe the earthquake effect in nepal. Reference n.2 gives value to the use of ambient vibration for dynamic monitoring. Finally reference n.3 explains the difference between infill masonry wall and rigid diaphgram respect the dynamic action 

- The conclusions are pooor and trivial

- The absence of any model to better understand the interaction between beam-column frames and infill masonry wall is not acceptable

- The benefit in shear resistance due to infill masonry walls is very obvious; by the way, the authors are requested to better explain that point also with the benefit's quantification due to the infill maosnry walls in tall building in varaying his height.

- Figure 8 is unclear and need clarifictaion and major support through text

- The incidence of symmetrical and asymmetrical plant  of the investigated building respect the effect of infill maosnry wall in tall building deserves more deepenings 

- Equations from 1 to 3 are trivial and more technique than scientific

- Fig 11 is without scientific explanation and sufficient clarification. By the way, why only one strut model and not two in diagonal?

- Fig 13 ( the second) is very of high intrerest and needs more thoughts

- The manuscipt has two fig 13, please correct.

Author Response

Dear Authors

The paper is potentially of high interest but needs major and important corrections. The requested improvement/corrections are reported in shynthetic way as following:

- The introduction needs to be improved and rewritted since that the influnce due to infill masonry walls in normal and taller framed building subjected to horizontal dynamic action is well known especially in Europe.

Author’s response:

The authors accepted the reviewer suggestion revised the manuscript introduction. The discussion concerning the masonry infill walls influence in the seismic performance of normal and taller buildings was improved.

- The introduction has to be improved with fruitfuill comparison than masonry bearing cracked part damaged in Nepal after the same earthquake in historic constructions

Author’s response:

The authors improved the discussion concerning the masonry infill walls influence in the seismic performance observed along the last major earthquakes in Europe and compared with the damages observed after the Gorkha earthquake. Regarding the comparison with the Historic constructions, the authors believe that could not be reasonable, since the masonry used in historic constructions have a different behaviour and contribution to the global seismic response of the building when compared with the masonry infill walls. A possible comparison between the damages observed in infilled RC structures and historic buildings is not possible to perform. However, the authors included a sentence in the introduction explaining that the large part of the collapses and extensive damages that occurred in the Gorkha earthquake were related to masonry buildings and historic constructions.

- The improve the introsduction also the following new reference are suggested:

1.  Brando G., Rapone D., Spacone E., O’Banion M. S., Olsen M. J., Barbosa A. R., Faggella M., Gigliotti R., Liberatore D., Russo S., Sorrentino L., Bose S., Stravidis A. (2017). Damage Reconnaissance of Unreinforced Masonry Bearing Wall Buildings After the 2015 Gorkha, Nepal, Earthquake. EARTHQUAKE SPECTRA, vol. 33, p. 243-273, ISSN: 8755-2930, doi: 10.1193/010817EQS009M

2. Russo S. (2016). Integrated assessment of monumental structures through ambient vibrations and ND tests: The case of Rialto Bridge. JOURNAL OF CULTURAL HERITAGE, vol. 19, p. 402-414, ISSN: 1296-2074, doi: 10.1016/j.culher.2016.01.008

3. Dal Cin A., Russo S. (2016). Annex and rigid diaphragm effects on the failure analysis and earthquake damages of historic churches. ENGINEERING FAILURE ANALYSIS, vol. 59, p. 122-139, ISSN: 1350-6307, doi: 10.1016/j.engfailanal.2015.09.010

The authors are not forced to use the new above proposed references in the introduction but they can concretely improve that chapter and the value of the references. By the way, with the aim to facilitate their use in the Introduction, the reference 1 gives the possibility to better describe the earthquake effect in nepal. Reference n.2 gives value to the use of ambient vibration for dynamic monitoring. Finally reference n.3 explains the difference between infill masonry wall and rigid diaphgram respect the dynamic action 

Author’s response:

The authors accepted the reviewer suggestions and included all the references.

- The conclusions are pooor and trivial

Author’s response:

The authors revised and re-written the manuscript conclusions.

- The absence of any model to better understand the interaction between beam-column frames and infill masonry wall is not acceptable

Author’s response:

The authors agree with the reviewer that is very important to understand the detailed interaction between beam-column joints and the infill frame. The strut model approach adopted in this research work is well accepted by the scientific community (as can be observed in the references [17-18]. This model allows to simulate the in-plane stiffness and strength contribution to the RC frame, and thus assess the real impact of the infill masonry walls in the building seismic response. Although, there is a need of detail modelling of the interaction between the infill masonry walls and the beam-column joints however there is a need of information of the masonry material and mechanical characteristics, which nowadays there is an absence of this amount of information regarding the Nepalese infill panels. Due to that and considering the methodology adopted by other researcher in other works based in the calibration of the building natural frequencies and assuming properly the modelling assumptions the authors believe that the strategy adopted is correct to assess the impact of the infill panels in the structure response, which is the objective of the present manuscript.

- The benefit in shear resistance due to infill masonry walls is very obvious; by the way, the authors are requested to better explain that point also with the benefit's quantification due to the infill maosnry walls in tall building in varaying his height.

Author’s response:

The authors believe that the contribution of the infill panels can be positive or negative depending on the several variables. Regarding the increase of the shear strength provided by the infill walls can be positive if the structural elements are well designed to support the shear loading transfer. If properly designed, the infill walls can globally improve the shear strength capacity of the building which can result in a better performance of the building when subjected to an earthquake.

- Figure 8 is unclear and need clarifictaion and major support through text

Author’s response:

The authors revised Figure 8 and improved the text that supports the Figure.

- The incidence of symmetrical and asymmetrical plant  of the investigated building respect the effect of infill maosnry wall in tall building deserves more deepenings 

Author’s response:

The authors agree with the reviewer and in fact the plan disposition of the infill walls could result in some asymmetry that could affected the structure and thus amplified the torsion effect. However, this could not be analyzed without also considering the vertical structural elements which also presents some asymmetry. Concerning the vertical disposition of the infill walls seems to present also some asymmetry, however the results from the survey damage assessment and numerical results did not revealed any concentration of the damages in the ground floor or any signal of soft-storey mechanism. A brief discussion was included in section 2.2.

- Equations from 1 to 3 are trivial and more technique than scientific

Author’s response:

The authors agree with the reviewer, however different approaches and methodologies can be adopted to determine the strut parameters. The authors believe that the inclusion of those equations could help the reader for a better understanding of the modelling process that was carried out.

- Fig 11 is without scientific explanation and sufficient clarification. By the way, why only one strut model and not two in diagonal?

Author’s response:

The authors agree with the reviewer and included a deeper scientific explanation of the one strut model. The purpose of the one strut model is to provide to the RC structure the stiffness and strength contribution of the infill panels. The strut is connected to the beam-column RC joints. It can be used two in diagonal (“X” disposition) but reducing the stiffness properties to half, and avoiding the possibility of assume higher stiffness than the expected.

- Fig 13 ( the second) is very of high intrerest and needs more thoughts

Author’s response:

The authors agree with the reviewer and included a deeper scientific discussion regarding the Figure 13.

- The manuscipt has two fig 13, please correct.

Author’s response:

The numbering of the Figures along the manuscript was revised.

Round 2

Reviewer 2 Report

Dear Author,

now the paper appears definitely improved.